# Linseed Silesia, Diverse Crops for Diverse Diets. New Solutions to Increase Dietary Lipids in Crop Species

**DOI:** 10.3390/foods10112675

**Published:** 2021-11-03

**Authors:** Magdalena Zuk, Jakub Szperlik, Jan Szopa

**Affiliations:** 1Faculty of Biotechnology, Wroclaw University, Przybyszewskiego 63/77, 51-148 Wrocław, Poland; jakub.szperlik@uwr.edu.pl; 2Linum Fundation, pl. Grunwaldzki 24A, 50-363 Wrocław, Poland; szopa@ibmb.uni.wroc.pl

**Keywords:** linseed, flax, Silesia, flax oil, flax oil stability, lipid peroxidation, polyunsaturated fatty acids, antioxidants

## Abstract

The aim of the work was to compare the new variety of oil flax (Silesia) with already cultivated varieties in terms of plant productivity, oil content, fatty acid composition and significant secondary metabolites. The analyzed linseed varieties are characterized by low (Linola), medium (Silesia) and high (Szafir) content of omega-3 fatty acids. Special attention was paid to the quality of the oil and the characteristics that determine its stability (reduction of susceptibility to oxidation). A number of antioxidant compounds of secondary metabolism (simple phenols, phenolic acids, flavonoids, tannins) were identified in the linseed oils. All of these compounds can affect lipid oxidation by a mechanism that attenuates initiating radicals such as hydroxyl or forms an oxidizing primary product such as peroxides. Chelation of metal ions may also be involved in lipid oxidation. We propose a mechanism that encompasses all these processes and facilitates understanding of the complex relationships between them. The general thesis is that the ratio of polyunsaturated fatty acids is associated with a better metabolic state of flaxseed, and thus with a higher nutritional value. In addition, we find a number of specialized secondary metabolites characteristic of the flax studied, which could be useful for chemotaxonomy.

## 1. Introduction

Flax (*Linum usitatissimum* L.), mainly linseed, has been cultivated for over two thousand years. The oilseed flax is primarily the source of valuable oils, of which the omega-3 fatty acids are most valued. Flax fibers (more commonly obtained from fibrous flax species) are used to make cloth (commonly referred to as flax). Fibrous flax and linseed are two different groups of varieties grown for the production of fiber and linseed oil, respectively. In this paper, we would like to focus on the oleaginous varieties of flax by presenting comparative studies on three flaxseed cultivars (grown in Poland) that differ in the biochemical composition of seed and oil. In recent decades, the use of flaxseed has focused on its use as a health-promoting component of the diet, supporting the prevention of civilization diseases [1,2,3].

The aim of this study was to evaluate the differentiation of plant productivity, oil content, fatty acid composition and essential secondary metabolites of phenylpropanoid and terpenoid metabolic pathways within three flax cultivars characterized by low (Linola), medium (Silesia) and high (Szafir) content of omega-3 fatty acids. The Silesia variety is the result of the induction of epigenetic changes (changes in DNA methylation) and the selection of plants with advantageous characteristics (ideal proportion of fatty acids, high resistance to field conditions and productivity).

Factors that favor the development of changes related to atherosclerosis are considered to be a high-fat diet, rich in saturated fatty acids (SFA) and cholesterol, and a low intake of fiber and polyunsaturated fatty acids (PUFA) [4]. These observations lead to the recommendation of a diet rich in PUFA and low in SFA as a factor in the prevention of many diseases and an ideal component of such a diet may be linseed oil. Among all presented polyunsaturated fatty acids (PUFA), α-linolenic acid (ALA) is widely regarded as the main beneficial component of flax for human health.

The fatty acids known to be essential for humans are α-linolenic acid, polyunsaturated fatty acid (PUFA) ω3 and linoleic acid ω6 PUFA. It is worth noting that from a nutritional perspective, deficiencies in both PUFAs are considered detrimental to human health. However, excessive levels of -6 polyunsaturated fatty acids and the very high ω6/ ω3 ratio in today’s Western diet promote the development of numerous diseases, including cardiovascular disease, cancer and inflammatory and autoimmune diseases, while elevated levels of ω3 PUFAs and the low ω6/ω3 ratio have a dampening effect. Therefore, a balance of 6 and ω3 fatty acids in the diet is very important for maintaining optimal growth and development of animals as well as for human health. Several studies have shown the health benefits of the diet of our ancestors, who had an estimated ratio of ALA/LA of 1:1, in contrast to the modern Western diet, which is 15–20:1. Ideally, the ratio of ω3:ω6 PUFA in the human diet would be restored to the original ratio of 1:1, which would reduce the incidence of chronic degenerative diseases [5]. The LA/ALA ratio has a higher value in evaluating the nutritional value of infant formula. This is because adult tissues synthesize fewer long-chain ω3-PUFAs than infant tissues, so the LA/ALA ratio in the diet does not have much impact on adults. A lower ω6/ω3 ratio in women with breast cancer was associated with a lower risk. The unfavorable regulation of cell and organ function determined by the interaction of ω3 and ω6 PUFAs could be due to inhibition or enhancement of inflammation. What is the molecular basis for this?

In a human cell model (aortic endothelial cells, HAEC), the production of reactive oxygen and nitrogen species (ROS) was measured after the cells were supplemented with various fatty acids. HAEC supplementation with polyunsaturated fatty acids resulted in lower ROS formation compared with cells supplemented with saturated or monounsaturated fatty acids. The ability of PUFA to scavenge free radicals therefore proves their regulatory role in oxidative homeostasis, reducing inflammation and thus the risk of cardiovascular disease [6].

Subsequently, studies in animal models showed that a low -ω3:ω6 PUFA ratio was associated with an increase in the expression of proinflammatory genes, proteins and cytokines related to lipid metabolism. Moreover, an optimal 1:1 PUFA dietary ratio of -ω6: ω3 in human and animal models significantly inhibited the expression of genes and proteins related to lipid metabolism, such as α-kinase-3-phosphoinositide, fatty acid transport protein-1 and peroxisome proliferator-activated receptor gamma (PPARγ), and suppressed the expression of inflammatory cytokines IL -1β, TNF-α and IL -6.

Thus, the optimal ratio of 1:1 -ω6: ω3 PUFA or the numerically lowest correction factor (CI)-3 calculated for the oil of Silesia cultivar had a beneficial effect on lipid metabolism and the inflammatory system, providing more energy and nutrients for high yield and homeostatic processes. The correction factor-3 ((CI = 1/2 3 + 6) − 6) indicates the percentage of deficiency (marked with a minus) or excess (marked with a plus) of alpha-linolenic acid in the oil of Linola, Silesia and Szafir varieties (respectively, −32, 5, +19.6, −4.5) that should be compensated for the optimal health-promoting value of the oil. In general, the UI reflects more comprehensively the proportion of FA with different degrees of unsaturation in the total composition FA of a species [7].

In addition, recently discovered molecules generated by the metabolism of PUFAs, such as arachidonic acid-derived epoxy-eicosatrienoic acid and lipoxins, as well as eicosapentaenoic acid-derived E-series resolvins, docosahexaenoic acid-derived D-series resolvins and finally protectins, appear to be essential to ensure timely resolution of inflammation and return to tissue homeostasis [8].

However, the undisputed disadvantage of oils rich in polyunsaturated fatty acids (PUFAs) is that they go rancid relatively quickly, to form chemicals that are harmful and even toxic to human health. Therefore, it is natural to search for or produce varieties characterized by a lower incidence of this unfavorable property of linseed oil.

One of the factors affecting the oxidative stability of vegetable oil is the composition of fatty acids, and the high content of polyunsaturated fatty acids such as linolenic acid is considered a pro-oxidant factor. In this work, we compare the susceptibility to oxidation of oils from three varieties of flax, characterized by different content of linolenic acid, in the simultaneous presence of antioxidant substances—the susceptibility of the oil to rancidity processes should be the result of the properties of all these substances in this situation [9].

Another goal we would like to achieve is to select the most effective combination of natural antioxidants that would protect the polyunsaturated fatty acids from oxidation already in the seed and later remain in the oil and increase its storage stability. To achieve this goal, it was also necessary to develop a model of the oxidative processes (including the involvement of antioxidants) that take place in linseed oil.

## 2. Materials and Methods

### 2.1. Plants Cultivation and Yield Determination

All studied cultivars were grown under field conditions in the same locations and under the same weather conditions (in three growing seasons 2018–2020) and in six climatically different locations across Poland (five experimental stations of the Research Center for Cultivar Testing (Polish name COBORU) and Wroclaw University of the Environmental and Life Sciences Experimental Station). Three of these experimental stations are located in the North of Poland—Wyczechy and Rychliki in the North and Krzyżewo in the North-East—a region of the country with favorable climatic conditions for flax cultivation (especially for flax fiber production). The remaining stations are located in the Southern part of Poland, where conditions are favorable for the cultivation of oil flax (seed production)—in South-Eastern Skołoszów and South-Western Głupczyce. (For details, see Experimental Stations (www.coboru.gov.pl (accessed on 4 October 2021.) In the COBORU experimental stations the tests for Distinctness, Uniformity and Stability (DUS) were carried out and the Value for Cultivation and Use (VCU) was demonstrated—prerequisites for registration in the National List and for granting the Plant Breeders’ rights for the Silesia variety (the status was obtained in March 2020) and would subsequently lead to registration on the EC Common Catalog. Moreover, similar tests and cultivations were carried out on experimental fields near Wroclaw (Wroclaw University of Environmental and Life Sciences test station). The described varieties showed some differentiation (3–27%) of yield-characterizing values (seed yield, 1000-seed weight, straw yield) between test stations—therefore, the average of all cultivation sites was taken as a basis for variety characteristics.

The overall field cultivation design was based on Randomized Complete Block Design with three replicates. Taking into account the edge effect (0.5 m), the middle rows of each plot were used to determine the following characteristics: seed yield (t/ha), weight of 1000 seeds (g), plant (straw) biomass (t/ha).

Weights were measured using a digital balance with an accuracy of 0.01 mg. Plots were harvested at maturity and seeds were dried to a uniform moisture content of 24–48 h at 60–70 °C. All measurements were given on dry weight basis.

### 2.2. Basic Nutrient Analysis

Seed protein, oil content and dietary fiber analysis were performed according to AOAC [10].

### 2.3. Determination of Main Polymers Content in the Straw and Fiber

The cellulose content was determined using the colorimetric method with the reagent anthrone, as described previously [11,12].

The determination of the total lignin content was performed using the acetyl bromide method, as described by Iiyama and Wallis [13].

The determination of the pectin and hemicellulose content was performed using the acetyl bromide method, as described previously [12,13].

### 2.4. Preparation of Oil from Seeds

First, 10 kg of flax seeds from field-grown plants was ground and transferred to an industrial worm gear oil press (Oil Press DD85G–IBG Monoforts Oekotec GmbH & Co., Berlin, Germany). This is a typical industrial method for cold pressing of oil. The obtained oil was stored at 4 °C under nitrogen (N_2_) until needed.

### 2.5. Determination of Fatty Acid Composition in Oil

Prior to GC analysis, the fatty acids were converted into their methyl esters (FAME). FAME analysis was carried out using an Agilent 7890A gas chromatograph with an FID detector on a DB-23 capillary column (60 m × 0.25 mm × 0.25 µm) suitable for the determination of fatty acid content and composition as described previously [14]. Each FA methyl ester was identified by its retention time and its quantity was calculated according to an internal standard. Each sample was esterified and measured in 3 replicates.

### 2.6. Extraction of Hydrophilic and Lipid-Soluble Components from Oil

The procedure for the extraction of phenolic compounds (phenolic acids, proanthocyanidin, hydrolysable tannins) and tocopherol, chlorophyll, lutein and plastochromanol-8 has been previously developed in our laboratory and described in detail in a previous publication [14].

### 2.7. UPLC-PDA-MS Analysis of Bioactive Components in Oil

Previously extracted phenolic and lipid-soluble compounds from the oil and seedcakes were measured using UPLC combined with two detectors PDA and MS (Waters Acquity UPLC System with a 2996 PDA detector and Waters Xevo QT of MS System mass spectrometer) using Acquity UPLC column BEH C18, 2.1; 100 mm × 1.7 μm and early selected for the separation of individual compounds mobile phases [14,15]. The MS spectra were recorded in ESI positive mode in the 50–800 Da range. The identity of the components was determined on the basis of retention time and UV and mass spectra of authentic standards.

### 2.8. Oil Stability Analysis

The peroxide value (PV) of the oil was determined by measuring the amount of iodine that was formed by the reaction of peroxides with the iodine ion. The peroxide value was measured as the mol/dm^3^ content of sodium thiosulfate (Na_2_S_2_O_3_). The conjugated diene and triene concentration was determined spectrophotometrically (234 nm) according to the published method [16]. Linoleic acid was used as a standard. The TBARS (reactive thiobarbituric acid) assay was used to measure the levels of the lipid oxidation product malonyldialdehyde (MDA) and other secondary lipid peroxidation products. As a result of the reaction, a pink pigment was produced. For more details, see [14].

### 2.9. Statistical Analysis

The data obtained were statistically analyzed using the t-test for independent samples and analysis of variance (ANOVA) by employing STATISTICA 10 software (StatSoft, Krakow, Poland). The results were statistically significant at *p* < 0.05.

## 3. Results

### 3.1. Crop Growth and Yield

All tested varieties were grown under field conditions in three growing seasons and in six different locations in Poland. The described varieties showed some differentiation (3–27%) of yield-characterizing values (seed yield, 1000-seed weight, straw yield) between test stations—therefore, the average of all cultivation sites was used as a basis for variety characteristics.

Field parameters showed significant differences in above-ground biomass (sum of stem, pod and seed weights) and seed yield, thus influencing the harvest index expressed as ratio of seed yield to above-ground biomass (H-index Linola—1.96/3.57 = 0.549; Szafir—2.08/3.82 = 0.544; Silesia—2.13/4.36 = 0.488). Plant height of Silesia variety was greater than the other tested flax varieties. Plant heights recorded in the field varied from 0.58 to 0.75 m for all the cultivars, with Linola (0.58 m) and Szafir (0.63 m) plants being the shortest compared to Silesia (0.75 m). Silesia was also superior to the other genotypes in yield, showing the highest yield in both total biomass (sum of straw and seed yield) and seeds (see Table 1 for details). It should also be noted that the largest mass of 1000 seeds is shown by plants of the Szafir variety.

The obtained data on chlorophyll content in the plant of Silesia cultivar were between 12% and 40% higher than in other plants during the whole plant development (Table 2). This is especially visible at the flowering stage (60 DAS, at the beginning of generative development. However, plants of the Silesia cultivar had the lowest harvest index (HI). The calculated HI values for Linola, Silesia and Szafir were 0.55, 0.49 and 0.54, respectively. Thus, although both grain and biomass yields differed among cultivars, variability was low at HI.

The likely physiologically based threshold for harvest index is about 0.6 [9,17]. Therefore, the approach to further improve the productivity of Silesia could focus on increasing the distribution of assimilates in seeds as a harvestable product. The number of seeds per capsule (data not shown) and thousand-seed weight also suggest that these parameters were influenced by the genetic traits of the cultivars studied.

The content of biopolymers in plant straw revealed considerable differences among taxa (Table 3). Silesia was characterized by the highest content of cellulose and hemicellulose and the lowest lignin content, which distinguished it from the other varieties. The fibers extracted from the straw showed significant differences in terms of lignin polymer. Silesia had the lowest content of this polymer.

The average oil and protein content of seeds (see Table 4) was only slightly influenced by variety, with the highest value recorded in Szafir variety with high linolenic acid content (45% and 23%, respectively). The oil content of linseed was reported to range from 34 to 45%, while the remaining linseed press cake contained high amounts of protein, ranging from 29.8 to 38.5% depending on the environmental parameters [18,19]. Fiber content was statistically higher for Szafir and Silesia compared to Linola.

The group of compounds with potential consumer toxic effects characteristic of flax are the cyanogenic glucosides. The presence of the monoglycosides linamarin and lotaustralin and the diglucosides linustatin and neolinustatin, the content of which varies during the growth and development of flax [19], was found in all the varieties studied. The total content of cyanogenic glucosides detected in the studied cultivars ranged from 11.88 mg/g (Silesia) to 14.49 mg/g for Szafir (data for mature seeds) and is a typical value for oil flax cultivars.

### 3.2. Fatty Acid Composition and Sensitivity to Oxidation

#### 3.2.1. Fatty Acid Composition in Oil Pressed from Mature Seeds

The most important flaxseed product with regard to its use in human and animal nutrition is the oil. Therefore, the quantitative and qualitative evaluation of the oil seems to be the key parameter characterizing the varieties of oil flax. The results of such analysis are presented in Table 5. Among the tested varieties, Silesia variety had the highest content of total fatty acids in the seeds (based on fresh weight)—45% and 20.5% more than Linola and Szafir, respectively.

#### 3.2.2. Flax Oil Sensitivity to Oxidation

In nature, fatty acids exist as mixtures of saturated fatty acids (SFA), monounsaturated fatty acids (MUFA) and polyunsaturated fatty acids (PUFA). The most frequently used parameter to assess the fatty acid composition is the unsaturation index (UI), which, calculated for the varieties Linola, Silesia and Szafir, showed the values of 161.8, 188.3 and 204.2, respectively. The unsaturation index (UI = 1 × (% monoenes) + 2 × (% dienoics) + 3 × (% trienoics)) indicates the effect of highly unsaturated fatty acids and does not ignore the effect of their low unsaturated counterparts [20]. The most abundant category in linseed is polyunsaturated fatty acids (PUFA) with the highest content of linoleic acid (18:2) in the Linola variety and alpha-linolenic acid (18:3) in Szafir, while the Silesia variety is characterized by an average content of both fatty acids. One of the factors affecting the oxidation stability of vegetable oil is the fatty acid composition of the oil. Primary oxidation of oil can be measured by determination of primary oxidation products such as conjugated dienes (CD) and conjugated trienes (CT) with peroxide value (PV). A secondary oxidation product such as malondialdehyde (MDA) also reacts with the polyunsaturated fatty acid composition of the oil. Thus, the content of both primary and secondary oxidation products reflects the polyunsaturated fatty acid content and hence the stability of the oil. Some limitation of this theory can be noted when considering the parameters of self-oxidation of oils after long-term storage at room temperature presented in this work. A higher content of ALA does not always lead to a higher oxidative lability of the oil, which is characterized by primary oxidation products such as PV, CD and CT, as well as secondary oxidation products such as the MDA content and the DPPH-scavenging activity of the oil expressed as IC-50. As shown in this study, the much higher content of ALA in Silesia variety leads to lower content of primary (PV) and secondary (MDA) oxidation products compared to Linola oil, which has a very low content of ALA. (See Figure 1 for details).

There are several reasons for this inconsistency. First, the oil stability parameters commonly used in the assessment of oil shelf life refer to oil as food and therefore provide limited data on the chemical kinetics of oxidation. Data from Differential Scanning Calorimetry (DSC) ranked oils according to their oxidative stability. The oxidation induction time at 20 °C, which mimics the onset of lipid oxidation, reflects the content of ALA. Linola oil had the latest oxidation onset time and the longest propagation time in all isotherms tested. The lowest value of these parameters was characteristic for the oil isolated from the Szafir variety. The suitable parameters for the oil of Silesia variety are between those of the Linola and Szafir varieties. Considering the time of oxidation onset as an indicator of the set point of primary product formation and the propagation time as the resistance of PUFA autooxidation, it can be assumed that both parameters correspond exactly to the ALA content. However, it should be noted that the kinetic parameters were derived from the heat-induced oxidation process, which does not necessarily reflect the stability of the oil at 20 °C.

Secondly, the oxidation process can generate fatty acid derivatives that affect its oxidation. In addition to hydroperoxides, conjugated fatty acids are among the most important autooxidation products, and their content is commonly used to determine the oxidation properties of fatty acids. The Szafir oil with the highest content of ALA has a lower content of conjugated dienes (CD), a comparable content of trienes and a higher MDA content compared to the Linola oil (Figure 1). The lowest values of the parameters CD and MDA, characteristic for the Silesia variety, could be due to the content of fatty acids and their ratio.

It is known that linoleic acid oxidizes 10–40 times faster than oleic acid and 2–4 times slower than alpha-linolenic acid [21]. For this reason, a formula was chosen to predict the oxidation index of the oil, taking into account the content and ratio of unsaturated acids. The calculated oxidation index for Linola, Silesia and Szafir oils was 0.88, 1.05 and 1.23, respectively. The result of oxidation of Linola oil with the addition of linolenic acids is in agreement with this. Increased MDA parameter of linolenic oil was observed after supplementation with linolenic acid (Figure 2).

Thus, the inconsistency of primary and secondary oxidation parameters with the content of fatty acids cannot be explained only by the ratio of unsaturated fatty acids. Therefore, the lowest value of the parameters CD and MDA in the case of the oil of the Silesia variety indicates the highest activity of antioxidants interrupting the lipid oxidation chain in this oil compared to the other oils studied.

### 3.3. Antioxidant Content in Oil

Many studies have shown inhibition of lipid peroxidation by hydrophilic compounds such as phenylpropanoid as well as by hydrophobic components such as tocopherols [14,22]. Seeds of the analyzed varieties of flax differ in the content of antioxidant compounds (for details, see Appendix A Appendix A). A number of antioxidant compounds of secondary metabolism were identified in the linseed oils (see Table 6).

Simple phenols, phenolic acids, flavonoids, condensed and hydrolysable tannins and lignins as well as tocopherols and carotenoids have been identified. All these compounds can influence lipid oxidation. Although the strong influence of phenylpropanoids on lipid stability is undisputed, they are present to a similar extent in the seeds of all three cultivars studied (the sum of the contents of the compounds between the cultivars does not exceed 18%), and we therefore assume that they cannot be considered as a distinguishing feature for the stability of the oils studied. An exception is individual compounds such as condensed and hydrolysable tannins, whose sum of the contents clearly distinguishes the oils (the sum of the contents differs by more than 40% between the varieties), and whose contents correlate with both the primary and secondary oxidation products.

## 4. Discussion

An important parameter determining the quality and agronomic usefulness of a plant variety is their productivity, especially in the aspect of raw materials obtained from a given plant (in the case of oilseed flax—valuable oil). We hypothesized that chlorophyll metabolism and thus photosynthesis contribute to the activation of lipid synthesis. Green leaves are characterized by the most active chlorophyll biosynthesis and photosynthetic efficiency important for plant growth [23]. Chlorophyll enabling the maintenance of photosynthetic capacity is also found in seeds, where the dominant function of photosynthesis is to increase the concentration of O_2_ to support the synthesis of ATP in the mitochondria [24]. A need for a high ATP content is required in seeds that synthesize a large amount of lipids because this biosynthesis requires the highest energy consumption compared to sugar or protein synthesis. This is confirmed by the observation that O_2_-supplied rapeseed improves ATP levels and lipid biosynthesis [25]. Additionally, acetyl-CoA carboxylase, a key enzyme in fatty acid synthesis, is activated after reduction, and the redox potential generated by photosynthesis is involved in this activation, therefore seed photosynthesis may contribute to the activation of lipid synthesis [26,27].

Plant biomass simply corresponds to photosynthetic activity or radiation use efficiency, while seed yield indicates the distribution of dry matter (assimilates) in seeds. Chlorophyll content and photosynthetic rate are positively correlated [23,24,25]. Maintaining adequate chlorophyll content in leaves throughout the growing season appears to be critical to achieving better growth and high yields in the cultivar Silesia. As we have already shown, the following three stages are crucial for flax development: vegetative growth, flowering, setting and growth. They are associated with significant changes in the profile of secondary metabolites such as flavones or cyanogenic glycosides [26,27,28].

The yield parameters were influenced by the genetic traits of the cultivars studied. It should be noted that the seed yields previously recorded for the variety Szafir grown in Poland [29] agreed with those presented in this paper, but the data concerning Linola appear in our case to be considerably higher than those published for this spring-sown variety in Central Italy [2]. Thus, certainly, the biometric parameters depend on geographical area, genotype and environmental conditions [18].

Linseed oil, high in the very important polyunsaturated fatty acids, appears to be of great importance due to the increasing consumer awareness of the strong relationship between food quality and health.

The undeniable advantage of linseed oil is its composition rich in polyunsaturated fatty acids.

On the other hand, PUFA-enriched oil easily oxidizes to form chemicals that are harmful and even toxic to human health [9,17]. However, measurements of oxidation parameters made on the oils described here differing in PUFA content, however, destroy the dogmas adopted so far. Silesia oil richer than Linola in unsaturated fatty acids is characterized by greater stability (less susceptibility to oxidation). In a study of oxidative stress in diabetic rats fed fructose, inconsistency was also found between parameters described as primary (CD) and secondary (TBARS) oxidation products [29].

An important observation is that oil from Silesia cultivar plants characterized by an average content of ALA has a lower content of conjugated dienes, a higher content of conjugated trienes and a lower MDA content compared to Linola oil and, in consequence, lower oxidation. On this basis, we suggest that conjugated fatty acids may delay the formation of the secondary oxidation product. Perhaps conjugated fatty acids are not only a marker of self-oxidation, but may also protect the fatty acid from further oxidation. Several publications describing the antioxidant properties of conjugated fatty acids are consistent with this conjecture. For example, foods containing pomegranate seed powder, which is very rich in CT, had lower TBARS values and reduced lipid oxidation by about 80% compared to the control group [18]. Protection against carcinogenesis by conjugated fatty acids has also been widely reported. For example, preclinical animal studies have shown that feeding CT inhibits colon tumor formation by modulating apoptosis and expression of PPARγ and p53 [19]. The anti-tumor mechanisms of fatty acids have been reported to include the peroxidation of membrane phospholipids, leading to inhibition of cell growth and apoptosis [20].

Oil of different plant origin (different genotype and chemotype) contains different types and amounts of compounds that can affect fatty acid oxidation. Additionally, seeking gentler methods of oil extraction and/or fortification of the oil with natural antioxidants increases its stability. For example, cold-pressed vegetable oils showed higher resistance to oxidation than other refined oils. An example is also transgenic flax which was enriched with water-soluble antioxidants. The transgenic vegetable oil showed an increased amount of phenols and a significant increase in the antioxidant capacity and stability of the oil [14,21]. Another example is that mixing linseed oil with cold-pressed Nigella sativa oil significantly improved its oxidative stability [22]. The main fatty acids of Nigella sativa were oleic, linoleic and alpha-linolenic acids with a content of 21.08%, 57.7% and 0.89%, respectively.

In general, lipid oxidation involves three phases: initiation, in which the unsaturated fatty acid reacts with a radical-forming initiator; propagation, in which the radical reacts with other fatty acids and spreads as a chain reaction; and termination, in which highly concentrated radicals interact with each other to form nonradical compounds [29] (see Figure 3).

In linoleic acid and alpha-linolenic acid, hydrogen splitting occurs at the methylene group (C11 and C11.14, respectively) between the double bonds, resulting in a fatty acid radical delocalized at carbons 9 to 13 and 9 to 16, respectively. The process of hydrogen splitting is essentially irreversible and basically results in a reaction of the fatty acid with oxygen. However, there are antioxidants that rapidly neutralize compounds that can trigger the formation of fatty acid radicals. This group of antioxidants includes enzymes such as superoxide dismutase, catalase and glutathione peroxidase, as well as other molecules of other types (e.g., proteins, tannins) that chelate ions such as iron and copper that inhibit or prevent the formation of fatty acid radicals [18]. Peroxide radicals are formed when the fatty acid radical reacts with oxygen, and the reaction is reversible. Scavenging antioxidants such as ascorbic acid, phenols and glutathione, which are hydrophilic, and tocopherols, which are lipophilic, can neutralize peroxide radicals and interrupt the propagation reaction, but they become free radicals themselves, albeit with less effect. The peroxide radicals can remove a hydrogen atom from the methylene group of other LA or ALA molecules to form a conjugated hydroperoxide product, or they can be further degraded by β-fragmentation. The natural antioxidants present in seeds/oils are effective hydrogen donors that reduce the peroxide radical to a fairly stable conjugated hydroperoxide that does not initiate or continue a radical chain reaction. The effectiveness of this reaction depends on the chemical structure and concentration of the antioxidants. Therefore, it can be assumed that the studied seeds of the Silesia variety contain compounds, the amount of which is most effective in protecting the oil from autooxidation. Taken together, this led to the assumption that the self-oxidation of lipids is accelerated by the increased content of polyunsaturated fatty acids and that the most reliable and efficient method of detecting fatty acid oxidation is scanning calorimetry; however, for a detailed analysis of the oxidation steps and the role of endogenous antioxidants in this process, it is necessary to measure both the primary and secondary oxidation products.

In general, compounds that inhibit lipid peroxidation act through a mechanism to quench initiating radicals such as hydroxyl, or to form an oxidizing primary product such as peroxides. Metal ion chelation may also be involved in lipid oxidation. We recently measured the levels of structurally different antioxidants (phenylpropanoids and terpenoids) in transgenic linseed oil. Oil enriched with phenolic compounds showed the highest antioxidant activity, and thus the oil’s stability. Therefore, it has been suggested that hydrophilic phenylpropanoids determine oil stability more than lipophilic isoprenoid compounds [14].

The hydrolysable tannins containing glucose as the central core esterified with gallic acid (gallotannins) and ellagic acid (ellagitanins). Proanthocyanidins, also known as condensed tannins, are macromolecules formed from the polymerization of flavanol subunits such as catechin and epicatechin and/or their derivatives (epigallocatechin, gallate-3-O-epicatechin). Several studies have shown that both hydrolysable tannin and condensed tannin affect lipid oxidation. One example is hydrolysable pomegranate tannin, which has high antioxidant potential and inhibits Fe^2+^-induced lipid peroxidation. Transition metal ions such as iron and copper are usually bound to proteins (ferritin or ceruloplasmin) and act as cofactors of antioxidant enzymes (catalase, superoxide dismutase, glutathione peroxidase). Individually, however, they can catalyze a radical reaction known as the Fenton reaction. Iron chelation by tannin has been shown to inhibit the Fenton reaction [30]. To better illustrate the processes described, a diagram was made to show them—see Figure 3.

Condensed tannins have an even higher antioxidant activity than hydrolysable tannins [31,32]. Tannins exhibit high antioxidant potential due to their high molecular weight and the high degree of hydroxylation of the aromatic components. Compared to other polyphenols such as flavanols, flavanols, hydroxycinnamic acids and simple phenolic acids, tannins are the most active [33,34,35]. Studies on the relationship between the structure and radical scavenging activity of numerous phenolic compounds identified in plants have shown that the extremely potent scavenging activity of tannins is due to their high degree of hydroxylation, glycosylation and methoxylation [36]. The ability of tannins to scavenge free radicals is essentially due to their ability to donate electrons to a free radical and produce a less harmful stable radical structure. A number of proanthocyanidins and hydrolysable tannins remove radicals such as superoxide radicals, hydroxyl radicals, peroxide and nitric oxide [36,37,38]. Moreover, a synergistic effect of the hydrolysable and condensed tannins was found. A 1:1 blend of tannins between hydrolysable and condensed tannins showed the best results in terms of both sensory quality and antioxidant activity and was suggested to be ideal for wine quality [31]. Based on these literature data and the results obtained, we suggest that the high stability of the oil of the Silesia variety is due to the high content of both types of tannins. A strong synergy of the action of both compounds may further enhance their antioxidant capacity and increase the protection of the oil against autoxidation.

Chemotaxonomy, and more recently chemophenetics, is an additional method of classifying plants based on a set of specialized secondary metabolites in taxa. This additional tool for plant identification helps to prevent adulteration and to plan and collect the cultivation of the desired plant material for a wide range of industrial applications, including medicine [32]. Because of the high efficiency of classification, the simplicity of analyzes and their low cost, further dynamic development of chemophenetics can be expected. As shown, in addition to fatty acids and tannins, linseed contains a number of other compounds that contribute to lipid protection. Some of them, such as carotenoids, tocopherols and lignans, are present in similar amounts in all flax varieties studied and are rather unsuitable for distinguishing plants on the basis of chemotaxonomy. The aim of this study was also to determine whether the content of carotenoids has potential in linseed taxonomy. The major carotenoids were lutein and β-carotene in all types of linseed oils. Despite slight differences in the content of both carotenoids between cultivars, the highest content was found throughout the growing season (three growth stages) in the above-ground plants, seeds and oil of the cultivar Silesia. Therefore, the data suggest that carotenoids are not compounds that can be used in the chemotaxonomy of flaxseed.

Tocopherols belong to a larger group of compounds, the tocochromanols [30]. The most important tocopherol was ɣ-tocopherol in all linseed oil cultivars, and its amount differed only slightly (within 14%) between cultivars, but the highest tocopherol content was found in the Silesia cultivar. We therefore suggest that tocopherols, like carotenoids, are useless for the taxonomy of linseed.

However, some metabolites are individual within the phenylpropanoid and terpenoid pathways and can serve as a basis for distinguishing plants by chemotaxonomy. One example is flavone, whose content in Silesia is 40–50% lower than in the other two cultivars; another example is coumaric acid, whose content in Linola is 30–40% lower than in the other cultivars. However, the most interesting are other tocochromanol representatives such as plastochromanol-8, as well as the total content of flavonoids. The amount of plastochromanol-8 or the sum of vitexin and proanthocyanidin corresponds exactly to the PUFA content. It can therefore be suggested that plastochromanol-8 and the sum of flavonoids, in addition to fatty acids, have an important chemotaxonomic value for the flax species.

The usefulness of plastochromanol-8 and tocopherols has already been investigated in terms of their taxonomic potential in flax. The data showed that these compounds have important chemotaxonomic value in addition to fatty acids [33].

The Silesia variety can also be seen as promising for current and future research directions. An example of a possible direction is the search for antiviral activity in secondary metabolites of the Silesia variety. The global transmission of COVID-19 and its high infectivity require the rapid development of appropriate prophylactic and therapeutic chemicals. Natural products can provide safe and cost-effective platforms for the discovery of effective and novel agents to treat viruses while minimizing side effects. Among the few compounds already identified such as catechins/polyphenols that inhibit the 3-chymotrypsin-like viral cysteine protease, epigallocatechin gallate has unusual binding sites for the spike protein, preventing the virus from attaching to host cells [6,7,8].

## 5. Conclusions

In this work, we investigated the effects of polyunsaturated fatty acid concentration and composition on the agronomy and chemistry of different flaxseed species. While significant differences in secondary metabolite content were detected in the seeds of the cultivars, leading to differential antioxidant stability of the oil, slight but positive differences were found in the agronomic parameters of the cultivars studied. These differences between cultivars provide a basis for breeding in which higher antioxidant content makes the polyunsaturated fatty acids of the oil less sensitive to environmental stress, including self-oxidation. Our results suggest that intervarietal variability in the ratio of linoleic/linolenic acid and the content of secondary compounds such as plastochromanol-8, as well as condensed and hydrolyzing tannins, which cause changes in the sensitivity of the oil to self-oxidation, could be appropriate parameters for breeders to initiate selection lines that maximize yield and nutritional value of flaxseed.

The cultivar Silesia exhibited the best resistance to PUFA autooxidation compared to the other cultivars, as it had the highest content of tannins and thus the highest yield of PUFA. The mechanisms responsible for the reduction in autooxidation of linseed oil (using Silesia oil as an example) associated with the increased tannin concentration are probably due to the large number of hydroxyl groups in these compounds and the ability to chelate transition metal ions such as iron and inhibit the Fenton reaction. However, the exact mechanism is not yet fully understood. Finally, a detailed study of the secondary compounds shows that the sum of flavonoids and plastochromanol-8, together with the ratio of linoleic and linolenic acids, can serve as markers for chemotaxonomy between species.

The highly sensitive cultivar Silesia, characterized by increased oil stability, favorable polyunsaturated fatty acid ratio for human nutrition and favorable tannin content, can be considered as a promising cultivar for future research and is useful for nutrition and medicine.

Overall, it can be concluded from this work that the ratio of polyunsaturated fatty acids is associated with better metabolic status of flaxseed and thus higher nutritional value and better health promotion. Specifically, we propose that, as in human and animal models, PUFA levels and the -ω6: ω3 fatty acid ratio also regulate the oxidation state in the plant and control lipid metabolism, and that PUFA derivatives are involved in the regulation of primary and secondary plant metabolism and are thus associated with a better metabolic state of flaxseed and hence with optimized plant growth and production.

## Figures and Tables

**Figure 1 foods-10-02675-f001:**
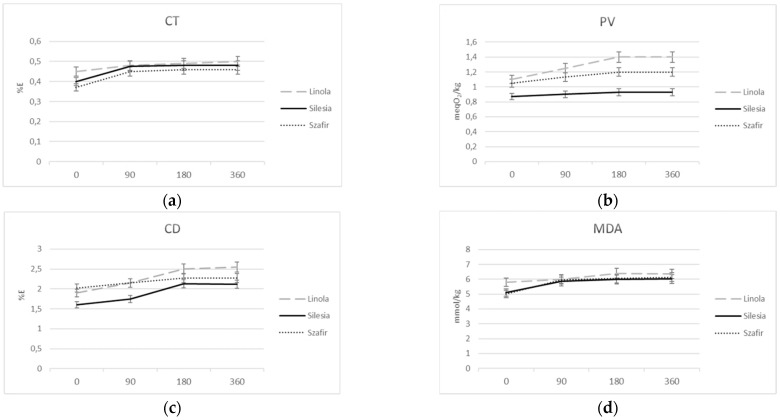
Fluctuation of primary and secondary oxidation products: (**a**)—conjugated trienes (CT) (**b**)—peroxide value (PV), (**c**)—conjugated dienes (CD),and secondary oxidation products: (**d**)—malondialdehyde (MDA) when the oil is stored in the refrigerator (4 °C) for 160 days. All results are statistically significant (*p* < 0.05).

**Figure 2 foods-10-02675-f002:**
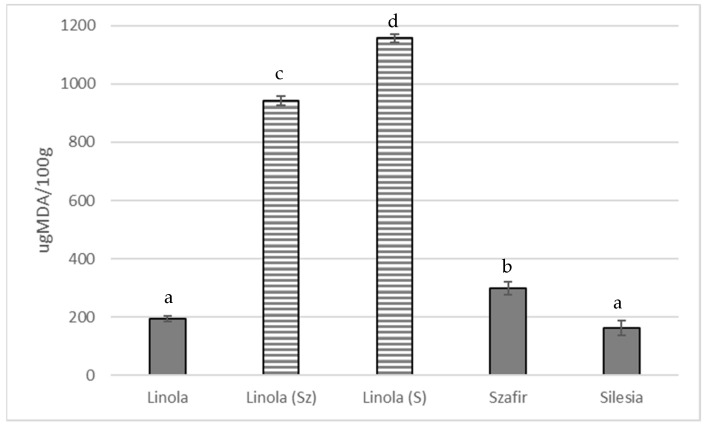
Analysis of MDA content after supplementation of Linola oil with linoleic acid to reach the levels characteristic for Szafir and Silesia oils (Linola (Sz) and Linola (S), respectively). The results typical for unattached Linola, Szafir and Silesia oils are given for comparison. All results are statistically significant (*p* < 0.05). Different superscripts imply significant differences.

**Figure 3 foods-10-02675-f003:**
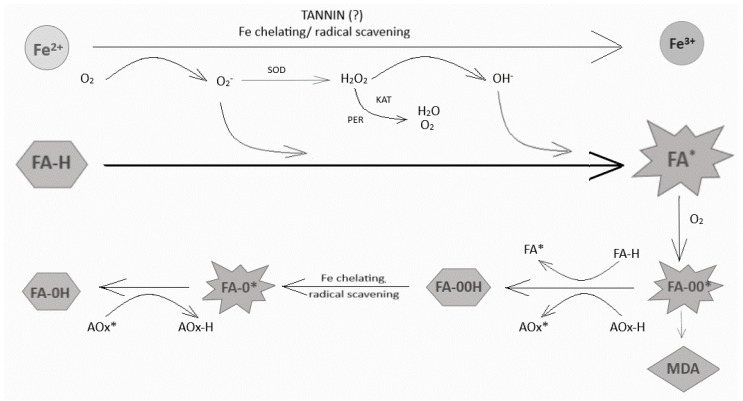
The process of lipid peroxidation (individual phases and the proposed involvement of antioxidants in combating this process). Lipid peroxidation phases: initiation, where the unsaturated fatty acid reacts with a radical-generating initiator; propagation, where the radical reacts with other fatty acids to propagate as a chain reaction; and termination, with highly concentrated radicals interacting with each other to obtain non-radical compounds. Hydrophilic compounds also play a role in inhibition of this process. SOD—superoxide dismutase, FA-H—unsaturated fatty acid, FA*—fatty acid radical, FA-OO*—fatty acid peroxide radical, AOx—antioxidant, MDA—malondialdehyde.

**Table 1 foods-10-02675-t001:** Yield parameters of the studied cultivars. Values for the vegetation periods studied and the average values of the varieties. (1000 seeds weight—1000 SW) All results are statistically significant (*p* < 0.05). Different superscripts imply significant differences.

	2018	2019	2020	Average
	Straw crop [t/ha]			
Linola	4.09 ± 0.10 ^a^	3.06 ± 0.13 ^a^	3.54 ± 0.12 ^a^	3.57 ± 0.13 ^a^
Szafir	4.33 ± 0.09 ^a^	3.29 ± 0.09 ^a^	3.84 ± 0.08 ^a^	3.82 ± 0.09 ^a^
Silesia	4.75 ± 0.09 ^b^	3.96 ± 0.10 ^b^	4.36 ± 0.09 ^a^	4.36 ± 0.09 ^b^
	Seed yield [t/ha]			
Linola	2.17 ± 0.03 ^a^	1.89 ± 0.03 ^a^	1.83 ± 0.03 ^b^	1,96 ± 0.03 ^a^
Szafir	2.25 ± 0.05 ^a^	1.92 ± 0.02 ^a^	2.08 ± 0.04 ^a^	2.08 ± 0.04 ^a^
Silesia	2.23 ± 0.03 ^a^	2.04 ± 0.04 ^b^	2.13 ± 0.03 ^a^	2.13 ± 0.03 ^a^
	1000 SW [g]			
Linola	5.6 ± 0.53 ^a^	5.7 ± 0.25 ^a^	5.8 ± 0.43 ^a^	5.7 ± 0.45 ^a^
Szafir	7.2 ± 0.63 ^b^	7.3 ± 0.33 ^b^	7.5 ± 0.52 ^b^	7.3 ± 0.63 ^b^
Silesia	5.8 ± 0.33 ^a^	5.9 ± 0.63 ^a^	6.0 ± 0.33 ^a^	5.9 ± 0.53 ^a^

**Table 2 foods-10-02675-t002:** Chlorophyll mg/gDW (sum chlorophyll a and b) in leaves at different growth stages. 30 DAS—vegetative growth, 60 DAS—flowering, 80 DAS—setting and growth of seed capsules. DAS—day after sowing. All results are statistically significant (*p* < 0.05) Different superscripts imply significant differences.

Variety	30 DAS	60 DAS	80 DAS
Linola	26.71 ± 1.25 ^a^	13.62 ± 0.55 ^a^	8.31 ± 0.33 ^a^
Szafir	21.82 ± 0.72 ^b^	12.59 ± 0.45 ^a^	11.64 ± 0.39 ^b^
Silesia	29.53 ± 1.06 ^c^	19.63 ± 0.63 ^b^	11.74 ± 0.42 ^b^

**Table 3 foods-10-02675-t003:** The content of the main biopolymers in plant straw and fibers. All results are statistically significant (*p* < 0.05). Different superscripts imply significant differences.

	Cellulose	Hemicellulose	Pectin	Lignin
The content of the main biopolymers (mg/gDW) in mature straw				
Linola	447 ± 19.2 ^a^	83.5 ± 10.67 ^a^	182.19 ± 13.98 ^a^	202.6 ± 53.7 ^a^
Szafir	507 ± 18.6 ^b^	96.9 ± 19.2 ^a^	179.65 ± 14.6 ^a^	194.6 ± 39.6 ^a^
Silesia	543 ± 12.5 ^c^	138.6 ± 12.81 ^b^	169.58 ± 17.73 ^a^	188.8 ± 24.5 ^a^
The content of the main biopolymers (mg/gDW) in fiber extracted from straw				
Linola	656 ± 19.5 ^a^	13.69 ± 3.99 ^a^	16.66 ± 2.70 ^a^	141.19 ± 16.13 ^a^
Szafir	664 ± 28.6 ^a^	12.89 ± 4.36 ^a^	16.44 ± 3.99 ^a^	138.26 ± 12.99 ^a^
Silesia	692 ± 35 ^a^	14.54 ± 3.43 ^a^	16.28 ± 2.01 ^a^	98.77 ± 7.04 ^a^

**Table 4 foods-10-02675-t004:** Seed protein, oil content and dietary fiber amount of linseed (%) depending on variety. All results are statistically significant (*p* < 0.05). Different superscripts imply significant differences.

Variety	Fats	Crude Protein	Dietary Fiber
Linola	41.3 ± 0.8 ^a^	20.78 ± 0.09 ^a^	4.23 ± 0.06 ^a^
Szafir	45.4 ± 0.5 ^b^	24.29 ± 0.12 ^b^	3.98 ± 0.06 ^a^
Silesia	44.1 ± 0.7 ^c^	24.20 ± 0.11 ^b^	4.08 ± 0.05 ^a^

**Table 5 foods-10-02675-t005:** Fatty acid composition in oil. FW—fresh weight of seeds. All results are statistically significant (*p* < 0.05). Different superscripts imply significant differences.

Variety	Linola	Szafir	Silesia
	Percentage of total fatty acids		
C16:0	6.78 ± 0.10 ^a^	6.01 ± 0.10 ^b^	4.92 ± 0.18 ^c^
C16:1	0.08 ± 0.02 ^a^	0.09 ± 0.01 ^a^	0.08 ± 0.02 ^a^
C16:2	0.06 ± 0.01 ^a^	0.06 ± 0.01 ^a^	0.04 ± 0.00 ^b^
C18:0	4.12 ± 0.14 ^a^	5.60 ± 0.06 ^b^	5.67 ± 0.11 ^c^
C18:1	19.54 ± 0.21 ^a^	22.81 ± 0.10 ^b^	20.74 ± 0.19 ^c^
C18:2	67.38 ± 0.44 ^a^	12.70 ± 0.07 ^b^	38.75 ± 0.32 ^c^
C18:3	2.40 ± 0.04 ^a^	51.88 ± 0.26 ^b^	29.91 ± 0.24 ^c^
C20:0	0.13 ± 0.02 ^a^	0.13 ± 0.03 ^a^	0.11 ± 0.01 ^a^
C20:1	0.09 ± 0.01 ^a^	0.12 ± 0.02 ^a^	0.09 ± 0.01 ^a^
C22:0	0.06 ± 0.01 ^a^	0.08 ± 0.01 ^b^	0.07 ± 0.00 ^c^
C22:1	0.05 ± 0.00 ^a^	0.05 ± 0.00 ^a^	0.04 ± 0.01 ^a^
	Sum of fatty acids mg/gFW		
∑ total	214.38 ± 5.84 ^a^	258.46 ± 5.66 ^b^	311.18 ± 4.95 ^c^
PUFA	149.72 ± 3.48 ^a^	167.07 ± 3.34 ^b^	213.78 ± 3.44 ^c^
MUFA	42.36 ± 0.55 ^a^	59.63 ± 0.36 ^b^	65.19 ± 0.52 ^c^
SFA	23.77 ± 0.26 ^a^	30.55 ± 0.30 ^b^	33.51 ± 0.39 ^c^

**Table 6 foods-10-02675-t006:** The content of antioxidants in oil depending on variety. All results are statistically significant (*p* < 0.05). Different superscripts imply significant differences.

Compounds (µg/100g FW)	Linola	Szafir	Silesia
Ferulic acid	0.79 ± 0.01 ^a^	1.39 ± 0.24 ^b^	2.15 ± 0.03 ^c^
Coumaric acid	0. 51 ± 0.01 ^a^	0.68 ± 0.03 ^a^	1.02 ± 0.01 ^b^
Caffeic acid	0.15 ± 0.05 ^a^	0.26 ± 0.02 ^b^	0.35 ± 0.006 ^c^
Chlorogenic acid	0.01 ± 0.001 ^a^	0.11 ± 0.01 ^a^	0.77 ± 0.018 ^b^
Vanillin	2.27 ± 0.01 ^a^	3.43 ± 0.72 ^b^	6.37 ± 0.001 ^c^
Syringic aldehyde	0.37 ± 0.01 ^a^	0.35 ± 0.03 ^a^	0. 53 ± 0.001 ^b^
Coniferyl aldehyde	0.47 ± 0.01 ^a^	1.58 ± 0.05 ^b^	2.74 ± 0.008 ^c^
Proanthocyanidin	0.15 ± 0.02 ^a^	0.26 ± 0.02 ^b^	0.27 ± 0.003 ^b^
Hydrolysable tannins	0.077 ± 0.05 ^a^	0.127 ± 0.05 ^b^	0.289 ± 0.011 ^c^
γ-Tocopherol	3579 ± 18.05 ^a^	3670 ± 15.85 ^b^	3628 ± 14.95 ^c^
Plastochromanol-8	45.92 ± 3.05 ^a^	48.80 ± 5.02 ^b^	52.84 ± 4.35 ^c^
Lutein	1.84 ± 0.15 ^a^	1.84 ± 0.14 ^a^	1.56 ± 0.14 ^b^

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
