# Peer review of "Linseed Silesia, Diverse Crops for Diverse Diets. New Solutions to Increase Dietary Lipids in Crop Species"

_foods, 2021, doi:10.3390/foods10112675_

Round 1
Reviewer 1 Report
Dear Authors,
In general, the paper is confusing in several sections. One of them is Material and methods, which is incomplete. Results section lacks a proper indication of statistical differences in tables and figures. Consequently, any interpretation and discussion of results is not properly supported. Therefore, major changes are necessary. Specific comments are described below.
Title of the paper is misleading. There are no innovative solutions presented in the paper. From a comprehensive perspective, the paper deals with
It is essential that the objective of the study is clearly stated in the paper. Please revise the objective of the study.
Line 15: flavonoids, and tannins)
Lines 19-21: It is not clear if the article is focusing on plant or human health. Please revise this paragraph.
Line 28: Linum usitatissimum should be italicized.
Lines 28-60: This paragraph must be rewritten because does not provide background information. Instead, it contains information that should be moved to the Material and Methods section.
So, why the characterization of these three species should be relevant? Why omega-3 has been considered as the main indicator to separate these three species? What is the importance of omega-3 rich oil from linseed? Maybe these questions assist you to create an introduction.
Results and Discussions sections: Since Results and Discussion are separated sections, the discussion presented the Results section must be moved the discussion section. For instance, lines 120-127 provide results about the crop growth and yield and the lines 141-148 provide an explanation for these results. Lines 120-127 must be preserved in Results section whereas lines 141-148 must be moved to the Discussion section. Other lines must be rewritten since results and discussion are merged into a single paragraph (such as in lines 188-192). Please check the Results section for similar occurrences.
Table 3 heading: biopolymers
Section 3.2.2 The description of the methods applied to obtain the results in this section is missing.
Section 3.3 This section only contains data about the phenolic profile of oils. Its heading must be revised to “Secondary compounds in oil”.
Chemotaxonomy of flax plants. Comprehensive statistical analysis is missing. Supporting the separation or grouping of linseed species with the data obtained with only argumentation is not enough. Please improve this section and Statistical analysis section.
Lines 340-343 What is the meaning of this statement?
Tables: Some vales are indicated using commas rather than a dot. Please check all tables.
Statistical analysis: Although indicated in the section 2.8. None of the tables or figures has letters to indicate the differences among average values. Please indicate the differences among treatments and make appropriate changes in the interpretation of results, discussion, and in the conclusion.
Please, give more attention to the organization of the paper. Captions of table are separated from their respective tables, some table has been split, caption to figure 1 is not correct… These inadequate formats do not comply with the instruction to authors/template in Foods.
Author Response
1.Title of the paper is misleading. There are no innovative solutions presented in the paper. From a comprehensive perspective, the paper deals with
In our opinion, the solutions presented are innovative - a new variety of flax (Silesia) has been generated, which has the characteristics of an "ideal source of lipids" for human nutrition. However, we understand that the reviewer sees it differently, so we have changed the word “innovative” to “new”.
- It is essential that the objective of the study is clearly stated in the paper. Please revise the objective of the study..
The objective of the study was stated in more detail - both in the abstract and in the introduction to the paper.
- Line 15: flavonoids, and tannins) -corrected
- Lines 19-21: It is not clear if the article is focusing on plant or human health. Please revise this paragraph.
This article focuses on improving the properties of flax seeds, more specifically the lipids they contain. From such seeds you can obtain a good oil and, above all, less prone to oxidation. Since flax oil can be considered a functional food (useful for nutrition and human health), a more distant goal is also to improve the quality of life and human health. This paragraph has been amended to improve its clarity.
5.Line 28: Linum usitatissimum should be italicized.
It was done.
6.Lines 28-60: This paragraph must be rewritten because does not provide background information. Instead, it contains information that should be moved to the Material and Methods section.
As suggested by the reviewer, some of the information has been moved to the Materials and Methods chapter. The introductory chapter has been thoroughly rewritten to emphasize the purpose of the work.
- Results and Discussions sections: Since Results and Discussion are separated sections, the discussion presented the Results section must be moved the discussion section.
We have rearranged these fragments (chapters).
- Table 3 heading: biopolymers – corrected
- Section 3.2.2 The description of the methods applied to obtain the results in this section is missing.
The absence of a description of the methods has been added. Thank you for paying attention - this text fragment must have "fallen out" of our hands during editing
10.Section 3.3 This section only contains data about the phenolic profile of oils. Its heading must be revised to “Secondary compounds in oil”.
The heading was changed.
- Chemotaxonomy of flax plants. Comprehensive statistical analysis is missing. Supporting the separation or grouping of linseed species with the data obtained with only argumentation is not enough. Please improve this section and Statistical analysis section.
We agree that the sample analysed (3 varieties) is too small to draw general conclusions. We have decided to move this section to Discussion - because the questions described here are hypotheses rather than results.
12.Lines 340-343 What is the meaning of this statement?
This sentence was changed
13.Tables: Some vales are indicated using commas rather than a dot. Please check all tables.
All tables were improved.
14.Statistical analysis: Although indicated in the section 2.8. None of the tables or figures has letters to indicate the differences among average values.
The determination of the statistical significance of the results has been added to the Tables and Figures
15.Please, give more attention to the organization of the paper. Captions of table are separated from their respective tables, some table has been split, caption to figure 1 is not correct… These inadequate formats do not comply with the instruction to authors/template in Foods.
I have the impression that some of these shifts are due to the reformatting of the document during the editing process and division into columns and pages. most of such situations have been corrected - the rest will be corrected in the final version of the manuscript.
Reviewer 2 Report
The aim of the study was to evaluate three flax cultivars (characterized by a different content of omega-3 fatty acids) in relation to: plant productivity, oil content, fatty acid composition and essential secondary metabolites. Particular attention was paid to the quality of the oil and to the characteristics that determine its stability (reduction of susceptibility to oxidation), for example the ratio between the quantity of linoleic and linolenic acid. Numerous antioxidant compounds have been identified in linseed oils such as simple phenols, phenolic acids, flavonoids, tannins, which can influence the oxidation of lipids. In this work the authors propose a mechanism that takes these processes into account, connecting them to each other. It was possible to identify the best cultivar, that is the one characterized by a lower self-degradation of polyunsaturated acids, which seems to be associated with the highest content of tannins. The authors underline how the identified cultivar can have useful applications in the food and medicine fields.

Author Response
Dear Reviewer,
Thank you very much for your valuable editorial comments - they have all been applied. Suggestions for correcting the list of literature were checked (e.g. abbreviations of journal names, missing DOI) and where it was justified supplemented/corrected.
Reviewer 3 Report
In the present manuscript, three flax cultivars were tested in order to evaluate the differentiation of plant productivity, oil content, fatty acid composition and essential secondary metabolites.
Introduction is minimal. The most of this section should be used in the Materials and Methods section (e.g. Lines 41-60). Selected parts of the discussion coulb used in the Introduction section (e.g. Lines 519-565)
Lines 73-75: what was the plot size and the number of plants per plot (plant density)?
What were the weather conditions? Provide a graph or Table.
What were the cultivation practices applied?
Provide statistics in Tables. Also, add a legend in Tables 3 and 5 above the Table.
Why the results from different locations are not presented? What is the point of using different locations in the scientific hypothesis?
Replace commas wiyth points in all the mean values presented in Tables 3, 5 and 6.
Revise the legend of Figure 1 according to the order of graphs presentation.
The results ection needs revisions since it contains references that should be used in the discussion section. The authors have to choose whether they present the results and discussion separately or combined.
Author Response
1.Introduction is minimal. The most of this section should be used in the Materials and Methods section (e.g. Lines 41-60). Selected parts of the discussion coulb used in the Introduction section (e.g. Lines 519-565)
The introduction and discussion have been thoroughly reworded at the suggestion of the reviewers.
2.Lines 73-75: what was the plot size and the number of plants per plot (plant density)?
What were the weather conditions? Provide a graph or Table.
What were the cultivation practices applied?
According to the guidelines for the registration process, the following rules applied to all crops:
Fertilizer was applied at 30 kg-ha-1 N in the form of ammonium nitrate 34%, 60 kg-ha-1 P2O5 in triple superphosphate 40% and 120 kg-ha-1 K2O in potassium salt 60%.
Sowing: row spacing 30 cm, sowing depth approx. 2-4 cm, 600 seeds per 1 m2.
We do not have accurate data on weather conditions during cultivation at all centres. The yield data presented is based on a report we received at the end of the cycle necessary to obtain rights to the variety. Of course, successive biochemical analyses were carried out (in our laboratory) in the following years of breeding. The presented results of yield analyses did not show significant differences depending on the place of cultivation (maximum 3-5%). A slightly larger variation was observed when comparing the growing seasons (these data are presented). Significantly larger differences were found between the analysed varieties.
- Provide statistics in Tables. Also, add a legend in Tables 3 and 5 above the Table.
Of course, our mistake:)
- Why the results from different locations are not presented? What is the point of using different locations in the scientific hypothesis?
Unfortunately, we do not have detailed data on growing conditions and at all centres / experimental stations. Most crops were grown as part of a 3-year registration survey (carried out by an external centre authorized to register new varieties). Thanks to the kindness of the staff at each station, we had access to small seed lots (not used for sowing the next season) each year. Often the seed lots reached us in the form of a mixture from several experimental stations.
Since the crop data are not our own (they are, of course, part of the registration studies we paid for and we have the right to use them), we present them as a characteristic of the plant material used.
Plants (seeds) from different growing areas were used for the analysis to eliminate the potential effect - the nature (type) of soil, small climatic differences in the quality of the seeds and the oil obtained from them. Of course, all three varieties were grown in each location. In our opinion, the use of such material gives credibility to the results obtained and the stability of the properties described.
- Replace commas wiyth points in all the mean values presented in Tables 3, 5 and 6.
It was done.
- Revise the legend of Figure 1 according to the order of graphs presentation.
It was corrected
9. The results section needs revisions since it contains references that should be used in the discussion section. The authors have to choose whether they present the results and discussion separately or combined.
The results and discussion have been thoroughly rewritten to avoid discussion elements in the description of the results.
Reviewer 4 Report
In the present work, Zuk et al. have evaluated the differentiation of plant productivity, oil content, fatty acid composition and essential secondary metabolites within three flax cultivars characterized by low (Linola), medium (Silesia) and high (Szafir) content of omega-3 fatty acids. The results are supported by the data and supply useful conclusion. There are some typewriting errors and some sentences are rambling. The conclusion section must be improved to better explain the obtained results and their potentiality: However, following suggestions are recommended:
-In the last paragraph of the introduction, the Author needs to clearly state the novelty of this paper together with future prospects of this study.
-Authors need to follow the journal format fully in the case of the Reference list. For example, Journal abbreviations, heading, and subheadings etc.
-In the result and discussion section, the author needs to pay more attention and validate their findings with recent previous results and compare if possible.
- The presentation of the Figure needs to be improved ( for instance figure 3). In some of them It is hard to read along the axis and also the corresponding title.
Author Response
1a) In the last paragraph of the introduction, the Author needs to clearly state the novelty of this paper together with future prospects of this study.
1b) In the result and discussion section, the author needs to pay more attention and validate their findings with recent previous results and compare if possible.
The Introduction, Result and Discussion have been thoroughly reworded at the suggestion of all reviewers. Particular attention has been paid to separating the purpose of the work and moving the elements of discussion from the Results to the Discussion (which facilitates the isolation of our own results) and allows them to be related to the achievements of other researchers.
2).Authors need to follow the journal format fully in the case of the Reference list. For example, Journal abbreviations, heading, and subheadings etc.
References and references to cited literature in the text have been thoroughly checked and corrected.-
3). The presentation of the Figure needs to be improved ( for instance figure 3). In some of them, It is hard to read along the axis and also the corresponding title.
Unfortunately, it is not possible to increase the font size in Figure 3 without affecting the readability of the diagram (with the drawing size required by the journal). The title of Figure 3 has been clarified.
Round 2
Reviewer 1 Report
Dear Authors,
Important changes have been made in the manuscript, but a major correction must be made. Statistical analysis. Indication of statistical analysis remains unclear. Statistical differences could be indicated with superscripted letter (abcd, for instance) in table and figures. Figure 1 remains without proper indication of differences.
Please check the appropriate statistical test (ANOVA) when analyzing the results. Please check other paper published in Foods for more information.
Author Response
As suggested by the reviewer, statistical differences were indicated witha superscript letter in all tables and figures.
In Figure 1, letter descriptions have been added next to the lines showing
the variability of the data (each data point is additionally provided with
an error bar)
Thank you for your time and valuable comments.
Reviewer 2 Report
The authors corrected the amount due and the article is now acceptable for publication. Some trivial formatting errors will be corrected when the proofs are ready.
Author Response
Thank you for your time and valuable comments on our manuscript.Reviewer 3 Report
The authors addressed all the comments. Therefore, I recommend the acceptance of the manuscript.
Author Response
Thank you for your time and valuable comments on our manuscript.This manuscript is a resubmission of an earlier submission. The following is a list of the peer review reports and author responses from that submission.